# The Landscape of microRNAs in βCell: Between Phenotype Maintenance and Protection

**DOI:** 10.3390/ijms22020803

**Published:** 2021-01-14

**Authors:** Giuseppina Emanuela Grieco, Noemi Brusco, Giada Licata, Daniela Fignani, Caterina Formichi, Laura Nigi, Guido Sebastiani, Francesco Dotta

**Affiliations:** 1Diabetes Unit, Department of Medicine, Surgery and Neurosciences, University of Siena, 53100 Siena, Italy; giusy.grieco.90@gmail.com (G.E.G.); noemibrusco91@gmail.com (N.B.); giadalicata.92@gmail.com (G.L.); dfignani@gmail.com (D.F.); catefo@libero.it (C.F.); launigi@gmail.com (L.N.); sebastianiguido@gmail.com (G.S.); 2Fondazione Umberto Di Mario, c/o Toscana Life Sciences, 53100 Siena, Italy; 3Tuscany Centre for Precision Medicine (CReMeP), 53100 Siena, Italy

**Keywords:** microRNAs, β cell identity, β cell function, β cell dedifferentiation, disallowed genes, β cell protection

## Abstract

Diabetes mellitus is a group of heterogeneous metabolic disorders characterized by chronic hyperglycaemia mainly due to pancreatic β cell death and/or dysfunction, caused by several types of stress such as glucotoxicity, lipotoxicity and inflammation. Different patho-physiological mechanisms driving β cell response to these stresses are tightly regulated by microRNAs (miRNAs), a class of negative regulators of gene expression, involved in pathogenic mechanisms occurring in diabetes and in its complications. In this review, we aim to shed light on the most important miRNAs regulating the maintenance and the robustness of β cell identity, as well as on those miRNAs involved in the pathogenesis of the two main forms of diabetes mellitus, i.e., type 1 and type 2 diabetes. Additionally, we acknowledge that the understanding of miRNAs-regulated molecular mechanisms is fundamental in order to develop specific and effective strategies based on miRNAs as therapeutic targets, employing innovative molecules.

## 1. Introduction

The etiopathogenesis of type 1 (T1D) and type 2 (T2D) diabetes mellitus is characterized by different mechanisms [1]; nevertheless, both are characterized by a consistent loss of functional β cell mass due to β cell destruction or dysfunction, leading to chronic hyperglycaemia. Until recently, the loss of β cell mass was almost exclusively attributed to apoptosis, while recent studies have shown that during metabolic and/or inflammatory stresses, β cells can undergo a loss of the mature phenotype, regressing to a progenitor-like state [2]. This process, known as dedifferentiation, represents a potential pathogenic mechanism, as the loss of the mature and differentiated phenotype of β cells negatively affects insulin synthesis and secretion. Although it is not yet completely clear how this mechanism affects β cell function, survival and proliferation, a series of studies have shown that dedifferentiated β cells are protected from apoptosis induced by both metabolic and/or inflammatory insults and by the attack of the immune system. Notably, based on this novel pathogenetic mechanism, it is possible to distinguish two types of loss of functional β cell mass: 1. caused by apoptosis and other mechanisms of cell death; 2. caused by dedifferentiation and/or other processes leading to β cell dysfunction.

In the interest of elucidating such mechanisms, several studies explored the role of microRNAs (miRNAs) in β cell function and plasticity. miRNAs are endogenous non-coding small RNAs, 19 to 24 nucleotides long, which negatively regulate gene expression through their specific binding to the 3’UTR region of messenger RNAs [3]. MiRNAs have been reported to regulate many cellular processes through the modulation of several signaling pathways components. Consequently, the alteration of miRNAs expression, reported in numerous diseases, contributes to defining a disease phenotype. Of note, several miRNAs have been reported to be involved in β cell differentiation, as well as in the maintenance of mature β cells phenotype and function [4,5,6]. Furthermore, specific miRNAs also play pivotal roles in compensatory mechanisms activated by β cells and triggered by several stress conditions such as glucotoxicity, lipotoxicity and/or inflammation, or by increased insulin demand due to insulin resistance phenomena [7,8].

In this review, we aim to report the latest advancements in the role of a set of miRNAs consistently demonstrated to be involved both in β cells’ phenotype maintenance and in the protection from typical diabetogenic stressors.

## 2. β Cell Identity and Function in Health and Disease

### 2.1. What Does Define the Path to βCell Identity?

Cell identity is determined by the expression of a set of genes that defines the cell-specific fingerprint. Such a set of genes represents the core identity of the cells and allows their peculiar functions within a tissue/organ.

For example, β cell function is guaranteed by the expression of key genes that promote: (i) the correct terminal cell differentiation program from pancreatic endocrine progenitors; (ii) the maintenance of signaling pathways and processes related to glucose-sensing machinery, insulin synthesis and secretion. Therefore, the loss of these genes during embryonic differentiation as well as in adulthood can lead to β cell dysfunction and a consequent impaired glucose metabolism.

During early and terminal β cell maturation programs, a cascade of molecular mechanisms regulated by several transcription factors (TFs) and signaling pathways leads to β cell maturation, thus directing the acquisition of a functional phenotype and promoting an accurate insulin release depending on peripheral demand [9,10].

β cell differentiation starts from a multipotent progenitor cell (MPC), whose specialization process is finely modulated by different TFs such as *GATA 4/6* (GATA Binding Protein 4/6), *FOXA2* (Forkhead Box A2), *PDX1* (Pancreatic And Duodenal Homeobox 1), *PTF1a* (Pancreas Associated Transcription Factor 1a), *MNX1* (Motor Neuron And Pancreas Homeobox 1), *SOX9* (SRY-Box Transcription Factor 9) and *NKX6.1* (Homeobox protein NKX6.1) that, depending on their expression levels, can induce MPCs to differentiate into a specific cell type belonging to one of the three major pancreatic compartments: exocrine, endocrine or ductal [11,12]. Further differentiation of endocrine progenitors occurs with the initial expression of *NGN3* (Neurogenin 3), followed by other TFs such as *ISL1* (ISL LIM Homeobox 1), *NEUROD1* (Neurogenic differentiation 1), *PAX6* (Paired Box 6), *MAFB* (MAFB ZIP Transcription Factor B), *NKX2.2* (NK2 Homeobox 2) and *RFX6* (Regulatory Factor X6). These TFs are directly activated by *NGN3* and are involved in the differentiation into several endocrine-pancreatic cell types (α, β, γ, and ε) [13].

The differentiation towards a more specific pancreatic endocrine cell identity is promoted by the controlled expression of two genes: *PAX4* (Paired Box 4) and *ARX* (Aristaless Related Homeobox). Indeed, the specific cross-talk among them results in the specific expression or inhibitory events that finally lead to the differentiation of islet progenitors into β/δ or α/γ identity [14]. Finally, the terminal definitive differentiation into mature β cell identity is achieved by the final activation of pivotal transcription factors, such as *NKX6.1*, *NKX2.2*, *PAX4/6*, *PDX1*, *MafA* [15], *FOXA2*, *NEUROD1* and *MNX1* [16].

### 2.2. How Do β Cells Maintain Their Identity?

In fully mature endocrine pancreatic cells, the presence of a defined set of TFs strongly contributes to the maintenance of the phenotype. Such a task is achieved both by promoting specific cellular functions and by repressing alternative transcriptional programs belonging to other cell types. This phenomenon is evident by ectopically activating α cell-specific TFs in a β cell context or vice versa; for instance, ectopic expression of *ARX* in adult β cells leads to the loss of β cell phenotype and acquisition of glucagon expression [17]. On the contrary, inactivation of *ARX* in mature α cells promotes their conversion into β-like cells. *PDX1* is a master transcriptional regulator that plays a key role both in pancreatic development and in adult β cell function. Indeed, in mature β cells, *PDX1* deletion leads to the loss of β cell identity. Intriguingly, upon *PDX1* deletion, an increase in α cell-specific genes was observed. As a matter of fact, in β cell context, *PDX1* specifically binds to *MafB* and glucagon genes promoters to suppress their activation, thus inhibiting a specific α cell transcriptional program [18]. In addition, *PDX1* represents a master regulator of multiple mature β cell functions, including the direct transcriptional activation of the insulin gene as well as of *MafA*. The latter controls additional downstream transcriptional factors and β cell functional effectors such as *GLUT2*, *NKX6.1*, *GLP-1R*, pro-hormone convertase-1/3 (*PC-1/3*) and pyruvate carboxylase [19].

Among TFs that strictly control β cell identity, Forkhead box protein O1 (*FoxO1*) certainly plays an important role. In mature β cells, FoxO1 is localized in the cytoplasm, then shuttling to the nucleus depending on the metabolic status of the cell, in order to maintain the expression of specific transcription factors such as *PDX1*, *NKX6.1* and *MafA*. However, chronic hyperglycaemia causes FoxO1 degradation, leading to decreased insulin content and upregulation of genes associated to the endocrine progenitor phenotype, such as *Vimentin* (mesenchymal marker), *Ngn3* (endocrine progenitor marker), *OCT4*, *Nanog* or *L-MYC* (stem cell markers) [20].

### 2.3. β Cell Phenotype Loss in Diabetes Mellitus

Recent studies demonstrated a high level of β cell plasticity during prolonged metabolic and/or inflammatory stress [21]. Such stressors can induce a loss of the mature β cell phenotype, leading to a regression to a progenitor stage (dedifferentiation) or to a transition toward another pancreatic endocrine cell type (transdifferentiation). Specifically, β cell phenotype loss in diabetes mellitus is characterized by (i) reduced expression of β cell-specific genes as well as of genes that regulate glucose-sensing machinery; (ii) hyperexpression of disallowed genes and of progenitor cell-enriched genes.

The first evidence of β cell phenotype loss was reported in rat models, in which prolonged exposure of pancreatic islets to hyperglycaemia caused a decrease in genes associated with glucose-induced insulin release as well as a reduction of several TFs involved in β cell development, differentiation and identity maintenance [22]. In 2012, Talchai et al. reported that β cell-specific *FoxO1*-KO mice subjected to metabolic stress showed a significant reduction in insulin-containing β cells while maintaining expression of specific markers of the endocrine lineage such as Chromogranin-A (*ChgA*) and Synaptophysin (*Syp*). Of note, *FoxO1*-deficient insulin-depleted β cells were Sox9-negative, Ngn3-negative, Pdx1-negative, Mafa-negative and ChgA-positive, thus indicating that they resembled endocrine pre-β-cells [20]. Furthermore, FoxO1 cytoplasm-to-nucleus translocation in β cells, previously observed in insulin resistance and in T2D animal models (i.e., GIRKO and *db/db* mice), was demonstrated to induce the expression of mesenchymal and stem-cell-associated genes (*Vimentin*, *Ngn3*, *Oct4*, *Nanog*, *L-Myc*), alongside loss of glucose-sensing ability and insulin secretion [23].

Importantly, increasing evidence of β cell dedifferentiation has also been reported in human islets in T2D context, thus resembling the molecular scenario previously observed in animal models. In 2013, Butler AE and colleagues analyzed pancreas sections from T2D donors and identified a population of dedifferentiated β cells defined by the positivity for the endocrine markers Syp or ChgA, but negative for insulin or for any other endocrine pancreatic hormone [24]. In addition, in pancreatic islets of T2D donors, Cinti et al. observed that dedifferentiated β cells (defined as insulin-/other hormones-negative and ChgA^+^/Syp^+^) showed the reduction and mislocalization of FOXO1 and NKX6.1 TFs and were enriched in ALDH1A3 (Aldehyde Dehydrogenase 1a3), previously demonstrated to be a marker of β cell dedifferentiation in T2D animal models [24,25,26]. Of interest, FOXO1 was ectopically found in glucagon-positive cells of T2D patients, thus potentially indicating that, in such a context, β cells become dedifferentiated and may undergo conversion to glucagon-positive cells [27].

Although dedifferentiation has been shown to be mainly triggered by gluco- and lipotoxicity, inflammatory conditions can also contribute to its induction. As a matter of fact, it has been observed that human pancreatic islets exposed to a prolonged low-grade inflammation (IL-1β, 1ng/mL) showed a reduction of β cell identity gene expression (e.g., *NKX6.1*, *GLUT2*, *FOXO1* and *MafA*), as well as an increase in *OCT4* and *NGN3* expression, indicating that inflammatory stress can also induce β cell dedifferentiation [28].

Indeed, in Non-Obese Diabetic (NOD) T1D animal model, Rui and colleagues observed potential clues of β cell dedifferentiation during the progression of the disease. In this study, authors identified a β cell subpopulation (defined “bottom β cells”) in the islets of NOD mice, characterized by reduced levels of genes associated to β cell function and increased levels of endocrine progenitor markers (e.g., *Ngn3*) and of stem cells factors (*Oct4*, *Sox2*, *Sox9* and *L-Myc*). Of major importance, dedifferentiated bottom β cells showed elevated levels of two factors associated with protection from immune system attack—*Pd-l1* (Programmed cell death-ligand 1) and *Qa2* (Qa lymphocyte Antigen 2)—resulting in a higher resistance of dedifferentiated bottom β cells to stress-induced inflammation (cytokines or lymphocytic infiltrates). These results suggest that dedifferentiation leads, on one hand, to β cell loss of function through the reduction of specific genes, while, on the other, to the presence of an inflammation-resistant phenotype [29]. As a matter of fact, it has been shown that in the NOD mouse, the specific deletion of the unfolded protein response (UPR) sensor *IRE1-α* prior to insulitis establishment led to the loss of β cell phenotype and was associated to a reduction of islet inflammation alongside with a relief of immune cells infiltration. Of importance, *Ire1-α*-deficient β cells were characterized by (*i*) reduced levels of typical β cell autoantigens; (*ii*) lower expression of MHC class I components; (*iii*) higher levels of immune inhibitory markers [30]. Collectively, these data indicate that β cell dedifferentiation may represent an escaping mechanism from the inflammatory insults during T1D progression.

Additional evidence of β cell dedifferentiation was reported based on the analysis of pancreata of donors with T1D, showing an increased frequency of ChgA^+^/hormone-negative cells, characterized by low levels of β-cell TFs [31,32]. Wasserfall et al. demonstrated that pancreatic islets of donors with recent-onset and long-standing T1D were characterized by low levels of insulin (protein), even though proinsulin mRNA was still clearly evident. The same islets were also characterized by a reduced expression of the enzyme PC1/3, with subsequent loss of β-cell function. These data suggest that the typical scenario of reduced functional β cell mass observed during T1D progression may be due, at least in part, to dedifferentiation phenomena [33].

As previously suggested, β cell phenotype loss in diabetes mellitus can also be the result of a transdifferentiation mechanism. During this process, fully mature endocrine cells are converted into another pancreatic endocrine cell type. As a matter of fact, in vitro primary human β cells can spontaneously undergo conversion into glucagon-producing α cells, putatively caused by the disruption of the islet architecture and loss of typical cell–cell and/or cell-matrix interactions [34]. Importantly, lentiviral-mediated knockdown of α cell-specific TF *ARX* in β cells inhibited β-to-α cell conversion, highlighting the critical role of a key transcription factor in the modulation of cell-specific phenotypes [34]. Of note, the maintenance of low expression levels of *ARX* (or other α cell-specific TFs) in β cells is pivotal in the retention of their identity. This concept is reinforced by additional data demonstrating that deletion of DNA methyltransferase 1 (*Dnmt1*) in β cells leads to de-repression of *Arx*, causing a consequent and progressive conversion of β cells into an α cell-like phenotype [35]. Of note, transdifferentiation events have been previously reported both in T1D and T2D [36,37,38], as well as in diabetic rodent models [39,40], in which prolonged hyperglycaemia could be responsible for β cell transdifferentiation [9].

Collectively, such evidence highlights the pronounced plasticity of the β cell phenotype, suggesting that dedifferentiated or transdifferentiated β cells can be induced to re-differentiate in order to restore the functional β cell mass.

## 3. microRNAs, β Cells and Disallowed Genes

As stated above, mature β cell identity is a tightly controlled status, whose homeostasis is guaranteed at multiple levels by the expression of transcriptional activators and/or repressors, modulation of epigenetic mechanisms and/or negative post-transcriptional regulation controlled by miRNAs.

Several studies focused on the role of miRNAs as potential modulators of glucose metabolism. Indeed, at present, several miRNAs have been clearly associated with the regulation of early and late stages of pancreas development, as well as of mature β cell identity and function.

MicroRNAs role in the pancreas and β cell differentiation and function was initially demonstrated through the inhibition of miRNAs maturation process in the pancreas or in β cells by specifically deleting the pre-miRNA processing enzyme *Dicer1* using *Pdx1*-CreER or RIP-CreER mouse models [41,42,43]. Conditional deletion of *Dicer1* in early pancreas development induced severe alterations in all endocrine pancreatic lineages, mostly due to the significant reduction of endocrine progenitors; these data indicate that miRNAs expression is required during endocrine pancreas organogenesis [43]. Furthermore, β cell-specific *Dicer1*-KO mouse models showed marked hyperglycaemia and glucose intolerance alongside a reduction of β cell mass and substantial alterations of islet architecture, thus demonstrating the crucial role of miRNAs in β cell maturation, identity and function [41]. Interestingly, it has been reported that specific deletion of *Dicer1* in β cells of 7- to 8-week-old mice (using Tamoxifen-induced conditional *Pdx1*-CreER mouse model) resulted in the upregulation of several disallowed genes [44]. In this study, Martinez-Sanchez and colleagues evaluated the expression of 14 selected disallowed genes (*C1qbp*, *Cd302*, *Cxcl12*, *Igfbp4*, *Lmo4*, *Maf*, *Oat*, *Pdgfra*, *Slc16a1*, *Smad3*, *Acot7*, *Ldha*, *Fcgrt* and *Ndgr2*) [45,46] in β cells derived from pancreatic islets of *Dicer*-KO mice. Among these 14 disallowed genes, six were significantly upregulated (*Fcgrt*, *Igfbp4*, *Maf*, *Oat*, *Pdgfra* and *Slc16a1*) and further shown to be directly affected by overall miRNAs reduction in *Dicer*-KO β cells. Interestingly, upregulation of these six disallowed genes correlated with the loss of insulin secretion at early stages of *Dicer* ablation and preceded β cell mass reduction. Since the loss of β cell identity is also characterized by increased expression of normally repressed or disallowed genes, it is possible to speculate that miRNAs dysregulation may significantly affect β cell phenotype by controlling the expression of those genes whose levels in β cells should be low or tightly monitored. As a matter of fact, during β cell phenotype loss in an in vitro model of human pancreatic islet cell dedifferentiation, we observed a general downregulation of β cell-expressed miRNAs, thus suggesting that the main alteration during β cell dedifferentiation is the loss of repression of a large set of target genes, which should be tightly controlled or repressed [47]. On the contrary, during β cell in vitro differentiation of human-induced pluripotent stem cells, the majority of miRNAs were upregulated, while only a small fraction was downregulated during the endocrine differentiation stages [48].

More specifically, inhibition of β cell-enriched miRNAs miR-200c, miR-182 and miR-125b has been demonstrated to increase the expression of *c-Maf* (c-Musculoaponeurotic fibrosarcoma oncogene homolog), which is involved in the regulation of glucagon synthesis in α cells and is considered a disallowed gene in β cells. In line with this, overexpression of the same miRNAs set in the αTC1 murine α cell line leads to a reduction in *c-Maf*, thus affecting α cell phenotype and function and demonstrating that multiple miRNAs may act in concert to control the expression of genes, which should be selectively repressed in β cells [49] (Figure 1). Another example derives from an additional set of β cell disallowed genes that encode for mitochondrial enzymes, co-factors and transporters. In fact, aberrant activation of these genes in β cells may lead to the induction of alternative metabolic pathways, thus interfering with glucose sensitivity and homeostasis. Among them, *MCT1* (MonoCarboxylate Transporter 1), which encodes for a lactate/pyruvate transporter, is of relevance. Indeed, low *MCT1* levels in β cells ensure that glucose-derived pyruvate is efficiently metabolized by mitochondria. It has been demonstrated that low levels of *MCT1* in β cells are maintained through the post-transcriptional repression exerted by miR-29a and miR-29b previously reported to be highly expressed in insulin-producing β cells [50,51,52], and by miR-124a, fundamental for pancreas development [53,54,55] (Figure 1).

## 4. microRNAs that Confer Robustness to β Cell Identity: Focus on miR-375, miR-7 and miR-204

Growth/survival- and proliferation-related genes represent an additional group of factors that should be tightly controlled in order to maintain a functional β cell phenotype [56,57]. As a matter of fact, cell cycle exit during endocrine progenitor differentiation facilitates terminal β cell fate specification. Among β cell-specific/enriched miRNAs, miR-375, miR-7 and miR-204 have been demonstrated as pivotal regulators of multiple β cell functions (Figure 2). Of importance, these miRNAs have been reported to be major determinants of the balance between β cell differentiation/phenotype maintenance and growth/proliferation.

### 4.1. miR-375

miR-375 is a pancreatic islet enriched miRNA. Indeed, it is prominently expressed in islet endocrine cells [54] and markedly involved in the maintenance of islet architecture by regulating β-to-α cell mass ratio [58]. Transcriptomic analysis of miR-375-KO mouse pancreatic islets showed that miR-375 regulates the expression of transcriptional repressors involved in the downregulation of growth genes. These data suggest that miR-375 regulates a cluster of genes involved in the repression of those factors that promote β cell growth/survival and proliferation. Indeed, Poy and colleagues showed that pancreatic β cell mass is decreased in miR-375-KO mice as a result of impaired proliferation; in contrast, pancreatic islets of obese mice (*ob/ob*) exhibited increased expression of miR-375, in consequence of a compensatory phenomenon occurring in pancreatic islets of this mouse model. Of note, genetic deletion of miR-375 in *ob/ob* mice (375/*ob*) profoundly reduced the proliferative/compensatory capacity of endocrine pancreas and resulted in a severe diabetic state.

In order to verify whether the overexpression of miR-375 may affect islet function and β cell proliferation also in a physiological context, Latreille and colleagues developed a non-obese/non-diabetic transgenic mouse model with selective miR-375 overexpression in β cells (resulting in a 2-fold overexpression vs. wild type β cells); however, no major change in β cell mass or function was observed, possibly due to the already elevated expression of miR-375 in β cells. However, an additional explanation may reside in the presence of redundant mechanisms selectively activated upon adaptation of the β cell to chronic overexpression of miR-375 in pancreatic islets of transgenic mice. Indeed, it has been shown that in vitro overexpression of miR-375 in primary β cells and in cell lines resulted in impaired insulin secretion, while its selective inhibition led to increased GSIS (Glucose Stimulated Insulin Secretion) through the modulation of multiple genes directly regulating insulin release (e.g., myotrophin (*MTPN*)) [58,59,60,61].

miR-375 expression is also of pivotal importance in pancreas organogenesis [62] as well as during β cell in vitro differentiation of human-induced pluripotent stem cells [48] and of embryonic stem cells. Indeed, when overexpressed in human embryonic stem cell-derived pancreatic progenitors, miR-375 induced cell cycle exit and promoted endocrine cell differentiation; of note, such an effect was mainly observed in combination with additional β cell enriched miRNAs (let-7g, let-7a, miR-200a and miR-127), demonstrating the role of multiple miRNAs network in the resulting phenotype [63]. Collectively, these data provide evidence that miR-375 is essential for normal glucose homeostasis, β cell differentiation and turnover, as well as for adaptive β cell expansion in response to increasing insulin demand.

### 4.2. miR-7

miR-7 is an evolutionarily highly conserved miRNA. miR-7 gene family is composed of three miRNAs, namely miR-7a-1, miR-7a-2 and miR-7b, whose transcription is particularly active and selective in β cells, thus resulting in elevated expression levels of this set of miRNAs. Latreille and colleagues showed that miR-7a negatively regulates GSIS by targeting regulators of insulin exocytosis. In addition, transgenic overexpression of miR-7a in murine pancreatic islets induced the reduction of *Ins1* and *Ins2* as well as of *Pdx1*, *Nkx6.1*, *MafA*, *Pax6*, *Gata6* and *Neurod1*, thus being involved in dedifferentiation mechanisms. In pancreatic islets of insulin-resistant mouse models (High-Fat Diet and *ob/ob*) and in pancreatic islets of mild/moderate T2D donors, characterized by increased β cell mass due to compensatory phenomena, miR-7 expression was reduced. On the contrary, in T2D rodent models characterized by severe hyperglycaemia and β cell alterations (*db/db*; non-obese Goto Kakizaki, GK diabetic rats) miR-7 was significantly hyperexpressed and linked to β cell phenotype loss [64,65].

The functional role of miR-7 has also been linked to the regulation of β cell proliferation, through targeting *mTOR* signaling pathway components [66]. Indeed, ex vivo inhibition of miR-7 in human and mouse pancreatic islets resulted in the activation of the *mTOR* pathway, thus inducing β cell replication [66].

Of relevance, miR-7 is also differentially expressed in hIPSCs [63] and in human embryonic stem cells (hESCs) during differentiation toward β cell fate [67]. Importantly, induced overexpression of miR-7 during hESCs differentiation improved generation of mature pancreatic insulin-producing cells and revealed the critical role of miR-7 in the modulation of β cell differentiation. In this context, induced miR-7 expression is paralleled by increased levels of typical endocrine pancreatic markers such as *PDX1* and *FOXA2* [67]. Overall, we can speculate that increased expression of miR-7 during β cell differentiation process may lead to a reduction in stem cell proliferation potential, alongside cell cycle exit and improved cell fate specification, thus remarkably phenocopying the effects of miR-375 in the same context.

### 4.3. miR-204

miR-204 is the most enriched miRNA in human β cells [68].Of note, this miRNA has been demonstrated to control the expression of *MafA* and of *GLP1R*. Xu et al. showed that miR-204 expression is regulated by thioredoxin-interacting protein (*TXNIP*), a cellular redox regulator involved in β cell physiology whose expression is increased in pancreatic islets in T2D. Indeed, *TXNIP* induces miR-204 expression, which in turn suppresses insulin production by directly targeting and downregulating *MafA*. This study suggests a *TXNIP*/miR-204/*MafA*/insulin pathway that could potentially contribute to diabetes progression [69]. Recently, Marzinotto et al. reported that miR-204 is also linked to in vitro hIPSC differentiation into insulin-producing cells. Indeed, it was demonstrated that miR-204 was progressively upregulated during hIPSCs maturation stages; however, authors did not find significant changes in *MafA* or *INS* mRNA levels upon up- or downregulation of miR-204 in both human islets and in the β cell line EndoC-βH1 [70].

Importantly, miR-204 has been shown to target *GLP1R* both in rodents and in humans. The reduction of miR-204 in β cells, both in miR-204-KO mice and in vitro antagomiR-204-treated cells, significantly induced *GLP1R* expression, resulting in the improvement of insulin secretion in response to exendin-4 and a better glucose control in streptozotocin-treated miR-204-KO vs. wild type mice [71]. Several studies in different contexts (e.g., cancer) suggested that miR-204 may be involved in the regulation of cell proliferation. Indeed, increased expression of miR-204 has been shown to inhibit proliferation in multiple cell lines, while its expression was significantly reduced in several tumoral tissues (showing a high proliferative capacity). In particular, a recent study demonstrated the anti-proliferative effects of miR-204 in *db/db* mouse islets, which were characterized by an increased cell proliferation upon genetic deletion of this miRNA, as well as by a reduced apoptosis induced by ER stress, consequently leading to an improvement of insulin secretion [72]. These data suggest that miR-204, in line with miR-7 and miR-375, may serve as a switch between proliferation and regulation of multiple β cell functions [72].

## 5. microRNAs in the Maintenance of β Cell Function: Protective and Compensatory Mechanisms in T2D

### 5.1. miR-24

The first study demonstrating a pivotal role of miRNAs in β cell protective mechanisms against lipotoxic or inflammatory stress in T2D was reported by Zhu and colleagues in 2013 [73], who demonstrated that miR-24, a miRNA highly expressed in pancreatic islets and enriched in β cells, was hyperexpressed in pancreatic islets isolated from *db/db* diabetic mice, as well as in MIN6 murine β cell line and in human pancreatic islets subjected to lipotoxic stress. Moreover, they demonstrated that miR-24 upregulation is able to induce *XBP1* and *ATF4* inhibition, both considered ER stress effectors, thus indicating a protective role of miR-24 by controlling the expression of a series of ER-related molecules.

More recently, upregulation of miR-24 in murine pancreatic islets was demonstrated to trigger a protective mechanism against thapsigargin-induced ER stress apoptosis, alongside the induction of β cell dedifferentiation [74]. Specifically, miR-24 upregulation was associated with increased expression of those genes linked to dedifferentiated β cells, such as *Ngn3*, *Oct4* and *Sox2*. In parallel, a specific downregulation of *Ins1* and *Ins2*, *MafA*, *NeuroD1* and *Pdx1* was observed as well. Of note, authors showed that miR-24 was able to protect MIN6 cells by directly targeting *Ire1α*, *Xbp1* and *Atf6* ER-stress effectors, thus protecting β cells from apoptosis while contributing to the loss of their identity. Collectively, these data demonstrate that miR-24 hyperexpression is able to prevent ER stress-induced β cell apoptosis while promoting β cell dedifferentiation and loss of identity, thus leading to β cell dysfunction.

The loss of β cell phenotype, along with reduction of ER stress, is of interest since it potentially indicates the need to alleviate ER workload by lowering insulin synthesis and processing. MiR-24 may represent a central node to coordinate β cells between survival (protection from apoptosis) and maintenance of a functional phenotype (dedifferentiation) (Figure 3).

### 5.2. miR-184

miR-184 is enriched in pancreatic islet β cells, where it is mainly involved in the regulation of compensatory processes in consequence of insulin resistance occurring in T2D. MiR-184 has been demonstrated to be downregulated both in T2D animal models and in human pancreatic islets from T2D donors [75,76,77].

In β cells, miR-184 negatively regulates the expression of its direct target gene *Ago2*, a pivotal component of RISC complex involved in the interaction between miRNAs and their specific target mRNAs. Indeed, in obese and in insulin-resistant mouse models, reduction of miR-184 induced *Ago2* upregulation leading to expansion of β cell mass. On the contrary, miR-184 overexpression downregulates *Ago2* expression [78]. Interestingly, *Ago2* upregulation secondary to miR-184 inhibition indirectly induces the enhancement of miR-375 effects on its target genes, which include those factors that repress proliferative genes in β cells. Overall, miR-184 downregulation leads to enhanced activation of miR-375, which, in turn, activates survival and proliferative cues.

Upstream signals that regulate miR-184 expression have been described by Martinez-Sanchez and colleagues, showing that nutrients and factors that modulate *AMPK* activity (including glucose) can modulate miR-184 expression in mice [79]. Indeed, they generated a β cell-specific *AMPK*-KO model and then evaluated miRNAs expression in pancreatic islets. Interestingly, miR-184 was the most downregulated miRNA, thus indicating a relationship with *AMPK* function. Interestingly, we recently demonstrated that miR-184 is also directly regulated by *NKX6.1*, previously shown to be altered in dedifferentiated β cells in T2D. Of note, we also demonstrated that reduced expression as well as nucleus-to-cytoplasm translocation of NKX6.1, typically observed in dedifferentiated β cells in T2D, leads to downregulation of miR-184. Furthermore, miR-184 reduction induced the upregulation of its target gene *CRTC1*, leading to β cells protection both from pro-inflammatory and lipotoxic stress-induced apoptosis [77]. Collectively, these results suggest that the observed reduction of miR-184-3p and the consequent upregulation of its target gene *CRTC1* in pancreatic islets of T2D donors is subsequent to the reduction/translocation of NKX6.1 occurring in dedifferentiating and/or dysfunctional β-cells in T2D, thus representing a link between protection and dedifferentiation [77]. Of note, the protective effect exerted by miR-184 against gluco-/lipotoxic and inflammatory stress was previously reported. Indeed, in 2013, Nesca and colleagues provided evidence that in vitro miR-184 inhibition in human islet cells induced (i) increased proliferation in response to prolactin exposure, (ii) reduced apoptosis rate following palmitate stress or exposure to pro-inflammatory cytokines [75]. Altogether, these data demonstrate the role of miR-184 in β cell response to compensatory cues and in β cell protection against lipotoxic and inflammatory stress (Figure 3).

### 5.3. miR-200 Family

MiR-200s is a highly conserved miRNAs family composed of five mature miRNAs, subdivided into two distinct gene clusters: miR-200c/miR-141 and miR-200a/miR-200b/miR-429. They are highly expressed in epithelial cells, where they prevent phenotypic transitions, such as epithelial-to-mesenchymal transitions (EMT), by targeting multiple genes involved in the exploitation of mesenchymal phenotype (e.g., *Zeb1* and *Zeb2* [80,81]). As a matter of fact, miR-200s have been defined as the guardians against epithelial phenotype loss.

In human pancreatic islets and in β cells, all components of the miR-200 family have been reported to be highly expressed [47,48], thus potentially controlling and maintaining mature phenotype. Additionally, miR-200s components have been described to be pivotal in the control of β cell apoptosis and survival [82]. In pancreatic islets of *db/db* T2D mouse model, miR-200s are significantly upregulated. Transgenic mice overexpressing miR-141/200c in β cells develop diabetes due to massive induction of apoptosis and consequent loss of β cell mass, leading to chronic hyperglycaemia and overt diabetes. On the other hand, genetic deletion of the two miRNA clusters (miR-141/miR-200a and miR-200b/c-3p/miR-429) confers protection against β cell apoptosis in three experimental models of apoptosis: the Akita mouse, the multiple low dose streptozotocin (STZ) model and the high-fat-diet-fed mouse treated with a single dose of STZ. In the same study, authors also demonstrated that the tumor suppressor *Trp53* is a direct target of miR-200 and that ablation of *Trp53* or of its downstream target *Bax* was able to prevent miR-200-induced β-cell apoptosis and, consequently, T2D development. Importantly, silencing of miR-200c with antagomiRs reduced the expression of pro-apoptotic genes activated by cytokines and associated with T2D in islets [83]. Nevertheless, complete loss of miR-200 family did not affect β cell function or glucose tolerance under basal conditions. Overall, miR-200s family members negatively regulate a conserved anti-apoptotic and stress-resistance network, which is reinforced upon ablation of those miRNAs which inhibit the expression of several components of such network. It is important to note that the miR-200 family has been shown to be significantly downregulated during the early stages of in vitro human β cell dedifferentiation [47] and that its potential downregulation along with a loss of β cell phenotype indicates a mechanism of dedifferentiation-induced protection through the contribution of specific miRNAs downregulation. Collectively, these data are in line with the novel view of the dedifferentiation process as a mechanism of β cell rescue from extracellular stimuli and signaling pathways leading to cell death (Figure 3).

## 6. Can We Protect or Restore β Cell Function by Mimicking or Antagonizing Key Groups of miRNAs?

The RNA-based therapeutic approach is an attractive strategy that may potentially lead to the specific correction of molecular defects or may regulate target cell functions by modulation of RNAs expression within the cell/tissue of interest.

Because of their multiple functions and involvement in the control of biological processes in β cells, miRNAs are considered optimal therapeutic targets. As a matter of fact, the modulation of specific miRNAs in pancreatic islets or in β cells can be used to achieve multiple goals:1.Generating in vitro functional β cells by modulating differentiation mechanisms of iPSCs and ESCs or by inducing transdifferentiation of other adult cell types into mature β cells. Substitution of highly expressed β cell miRNAs and/or silencing of dominant non-β cell miRNAs such as liver miR-122 [84] or neuronal miR-124 [85] have been demonstrated to affect the ability of multipotent stem cells to differentiate into insulin-positive cells. Indeed, virus-mediated overexpression of miR-375 in human skin fibroblast-derived iPSCs was sufficient to trigger their differentiation into insulin-expressing cells and to allow glucose-dependent insulin secretion in vitro [86]. Overexpression of miR-186 and miR-375 by chemical transfection of human iPSCs promoted the generation of islet-like cell clusters and induced the expression of β cell-specific markers [87].2.Protecting β cells from gluco-lipotoxic and/or inflammatory stress. A recent work by Zhu and colleagues reported that microRNA miR-24, which is upregulated in pancreatic islets of diabetic *db/db* mice [73] as well as in islet cells subjected to palmitate-induced lipotoxicity [74], directly binds and regulates the expression of *Ire1α* in MIN6 murine β cell line (see above). Importantly, downregulation of *Ire1α* secondary to the overexpression of miR-24 was able to protect β cells from both palmitate- and thapsigargin- induced apoptosis [74]. Accordingly, it was demonstrated that Ire1α deletion restricted to β cells was able to inhibit β cell apoptosis, thus preventing disease onset in NOD mice [30]. Therefore, it is conceivable that the regulation of *Ire1α -XBP1* by miR-24 is part of the molecular mechanisms involved in β cell protection from inflammatory-stress induced apoptosis. Altogether, these data indicate that modulation of miR-24 expression could be essential to protect β cells from apoptosis induced by different types of ER stress e.g., inflammatory and lipotoxic, although impairing β cell identity and function. Importantly, a peculiar miRNA that could link protection from stress and proliferative/compensatory events is miR-184, an interesting potential therapeutic target in diabetes mellitus. As a matter of fact, miR-184 downregulation protects β cells from both gluco/lipotoxicity- and inflammation- induced apoptosis [75,77].3.Proliferation and/or regeneration of β cell. Interestingly, miR-184 inhibition leads to increased β cell mass, mainly by enhancing β cell proliferative capacity, as described by Tattikota et al., (see above) [78]. Finally, miR-7 is an additional candidate to be targeted in order to generate a therapy aiming at inducing β cell proliferation [66].

Overall, the role of several miRNAs has been well described in different protective mechanisms that could be potentially targeted in order to develop an efficient therapeutic strategy against both T1D and T2D.

With this ambitious goal, a series of efficient approaches aimed at specifically increasing or decreasing miRNAs expression and function have been developed, since the modulation of small RNAs in cells has become a standardized procedure through transient or stable transfection or viral transduction of pri-miRNAs, pre-miRNAs, mature miRNAs, small interfering RNAs (siRNA), short hairpin RNAs or antagomiRs [88,89].

Inhibition of a specific miRNA can be mainly achieved through two different strategies: antagomiRs and miRNA sponges.

AntagomiRs were the first miRNA inhibitors that were shown to be fully functional in mammals [90]. They are synthetic antisense oligonucleotides containing 2-O-methyl-modified ribose sugars and terminal phosphorothioates and are conjugated at the 3′ end with a cholesterol molecule, which facilitates their delivery into cells by associating with lipoproteins and binding to cell membrane receptors [91,92]. Upon cell entry, antagomiRs are localized in the cytoplasm and do not affect pre-miRNA levels, thus targeting exclusively mature miRNAs through high-affinity base pairing, which prevents miRNAs binding to their target transcripts. Importantly, a Locked Nucleic Acid (LNA)-based strategy has already been developed in order to inhibit miR-24 expression in HFD-fed mice. Interestingly, this therapy was effective to reduce fat accumulation in both liver and plasma of this animal model without any adverse effects on other tissues/organs [93].

MiRNA sponges are tandem-repeated miRNA target sequences located after a reporter gene and work as miRNA decoys avoiding its binding to a target Mrna [94]. MiRNA sponges can be naturally found within long non-coding RNA sequences; nevertheless, they can also be easily synthetically produced as plasmid or viral vectors containing tandem arrayed miRNA binding sites, separated from each other by small nucleotide sequence spacers and inserted into a 3′UTR portion of a reporter gene driven by an RNA polymerase II promoter [95].

On the other hand, mimicking strategies mainly rely on double-stranded non-natural small RNA molecules, whose function is to mimic and replace natural transcribed miRNAs [96]. Importantly, miRNA mimics can be fully designed and chemically modified based on the target tissue and different experimental applications [97,98].

However, even though multiple strategies have already been developed to specifically increase or decrease miRNAs expression, selective delivery of synthetic RNA molecules to a specific cell of interest, without affecting surrounding tissue, appears quite challenging. Indeed, off-target cell delivery of these molecules represent the main issue of RNA-based therapeutics. As a matter of fact, these molecules have been reported to induce different side effects, such as platelet activation and consequent thrombus formation in specific contexts [99]. In order to overcome this problem, a series of strategies have been developed to narrow as much as possible the delivery to a set of target cells [100,101]. Among them, the most important are certainly aptamers [102], which are short single-stranded(ss) DNA or RNA molecules selected for the binding to a specific target [103], representing valid vehicles for miRNA-drugs delivery [104]. Furthermore, aptamers can be subjected to specific modifications, thus potentially avoiding side-effects secondary to the lack of tissue-specificity while maintaining their therapeutic efficacy.

An issue to be solved is represented by compounds conjugated with Antagomirs.

For instance, PS-conjugated Antisense Oligonucleotide drugs have been demonstrated to have important side effects such as complement activation and prolongation of aPTT (activated partial thromboplastin time). In particular, the immunostimulatory effects of PS-conjugated Antisense Oligonucleotide drugs have been shown to be sequence-dependent and mainly associated with the presence of CpG motifs [105].

The administration route of miRNA-drugs is an additional aspect that remains to be standardized. Indeed, different administration strategies of specific miRNA-drugs could not work in a particular organ or tissue (e.g., due to the action of several enzymes) [106].

For all these reasons, although miRNAdrugs remain a promising tool for the treatment of several disorders, including T1D and T2D, we are currently far from the clinical administration of these compounds to human subjects.

## 7. Concluding Remarks

In recent years, compelling evidence has shown that microRNAs are major regulators of most mechanisms of β cell development, differentiation, maintenance of identity and function. Given their involvement in all these processes and taking into account the recent advances in technologies for in vivo miRNA modulation, specific sets of miRNAs may represent novel therapeutic targets both in T1D and in T2D.

## Figures and Tables

**Figure 1 ijms-22-00803-f001:**
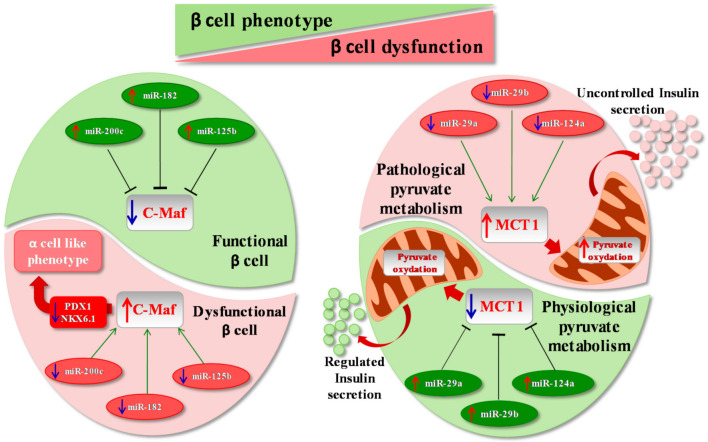
MicroRNAs and disallowed genes. The inhibition of disallowed genes exerted by several microRNAs (miRNAs) is essential in order to maintain functional β cell identity. (Left panel) Indeed, for instance, physiological expression of three miRNAs (miR-200c, miR-182 and miR-125b) is able to inhibit expression of *c-Maf*, a gene usually absent in mature β cells but enriched in α cells (*green half of cell*). On the contrary, downregulation of miR-200c, miR-182 and miR-125b leads to pathological hyperexpression of *c-Maf* in β cells, leading to a reduction of essential β cell genes and transcription factors, thus inducing β cells to switch to an α cell-like phenotype (*pink half of cell*). Importantly, loss of β cell identity also leads to β cell dysfunction and metabolic alterations. (Right panel) A particular disallowed gene, *MCT1*, physiologically inhibited by miR-29a/b and miR-124a, is involved in the regulation of insulin secretion through the modulation of pyruvate metabolism (*green half of cell*). In addition, reduction of the previously cited miRNAs leads to *MCT1* hyperexpression in β cell, thus inducing pyruvate oxidation and uncontrolled insulin secretion (*pink half of the cell*).

**Figure 2 ijms-22-00803-f002:**
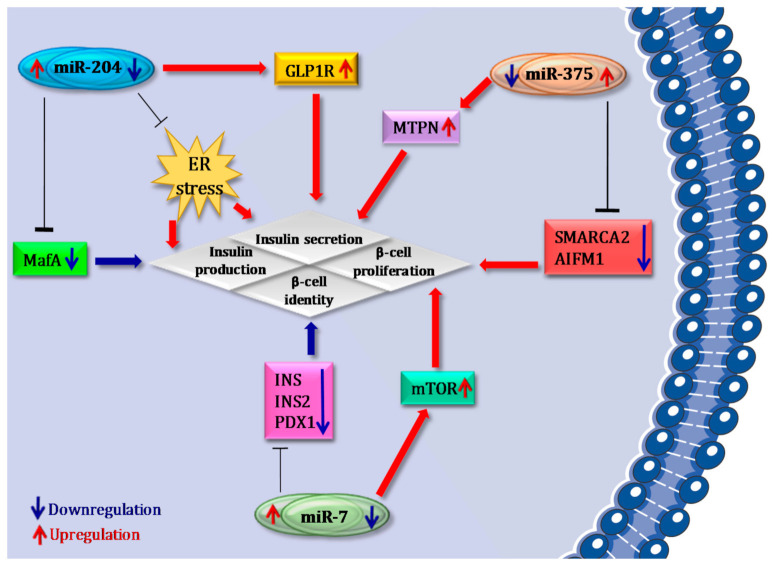
MicroRNAs in β cell identity and function. Several miRNAs regulate β cell identity and function. Hyperexpression of miR-204 induced by *TXNIP* causes suppression of insulin production by directly targeting and downregulating *MafA*; on the other hand, reduction of miR-204 induces GLP1R upregulation, thus resulting in an improvement in insulin secretion. Genetic deletion of miR-204 protects β cells from ER stress-induced apoptosis, thus leading to improved insulin secretion. Importantly, miR-375 hyperexpression is able to stimulate β cell proliferation, mainly by inhibiting anti-proliferative genes such as *Smarca2* and *Aifm1*, while selective inhibition of miR-375 leads to upregulation of *MTPN*, thus improving glucose-stimulated insulin secretion. Another miRNA involved in β cell proliferation is miR-7. Indeed, its downregulation is able to induce hyperexpression and the subsequent activation of *mTOR* pathway, leading to β cell replication. On the contrary, transgenic hyperexpression of miR-7 in murine pancreatic islets leads to loss of β cell identity by inhibiting the expression of *Ins1*, *Ins2*, *Pdx1* and other genes involved in the maintenance of β cell identity.

**Figure 3 ijms-22-00803-f003:**
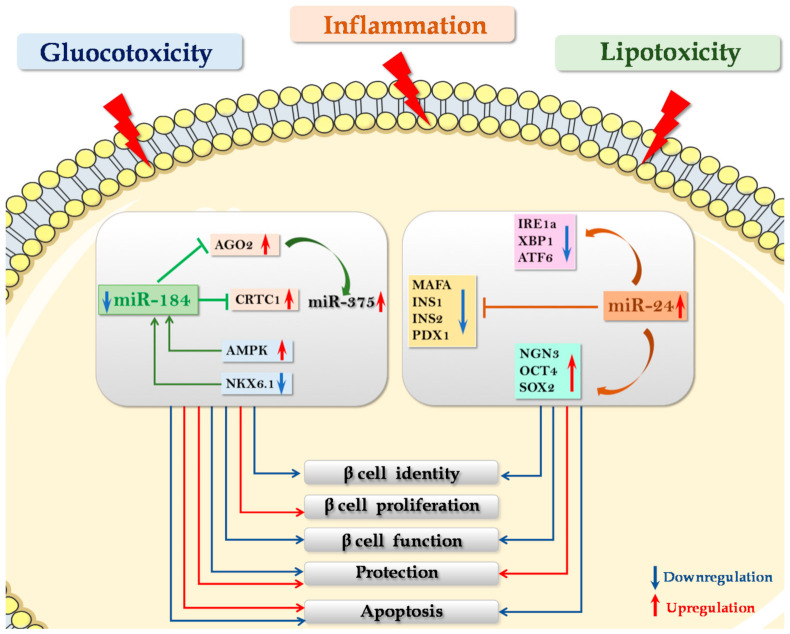
MicroRNAs as regulators of β cell compensation and protection. Metabolic (glucotoxicity and lipotoxicity) and inflammatory stress modulates miR-184, miR-375 and miR-24 expression in β cells. Specifically, miR-184 downregulation induces *Ago2* upregulation, leading to expansion of β cell mass. MiR-184-mediated *Ago2* upregulation indirectly enhances miR-375 effects on its target genes, which include factors that repress activation of proliferative genes in β cells. miR-184 downregulation leads to enhanced activation of miR-375, which, in turn, activates cell survival and proliferation. Nutrients and factors that modulate *AMPK* activity (including glucose) can modulate miR-184 expression. Inhibition of miR-184 in human islets increases proliferation and reduces apoptosis following lipotoxic and inflammatory stress. *NKX6.1* directly regulates miR-184 promoter; consequently, the reduced expression of *NKX6.1* leads to downregulation of miR-184, which in turn causes overexpression of *CRTC1* with subsequent β cell protection from apoptosis. MiR-24 upregulation inhibits ER stress-related target genes (*IRE1**α*, *XBP1* and *ATF6*). Upregulation of miR-24 induces genes typically expressed in dedifferentiated β cells, such as *Ngn3*, *Oct4* and *Sox2* together with a downregulation of *Ins1* and *Ins2*, *MafA*, *NeuroD1* and *Pdx1*, thus leading to the loss of β cell identity and function and to increased protection from apoptosis. The red arrow indicates upregulation/activation, and the blue arrow indicates downregulation/inhibition.

## Data Availability

Data (figures) always available under citation.

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
