# Peer review of "The Landscape of microRNAs in βCell: Between Phenotype Maintenance and Protection"

_ijms, 2021, doi:10.3390/ijms22020803_

Round 1
Reviewer 1 Report
This review is comprehensive and provides an excellent overview of the field. I only have a few comments:
- The term 'temporary' used in line 38 may mistakingly lead to reader to think that the loss of beta-cell function will at some point return to normal, but this is clearly not the case as the dedifferrentiation or dysfunction of these 'silent' beta-cells seem chronic. Consider to rephrase the term.
- In generel, the authors do not differentiate between which miRNAs that they describe are mainly relevant for type 1 diabetes and which for type 2 diabetes and which are for both. Maybe it would strenghten the manuscript if the authors put slightly more attention as to describe which specific miRNAs are believed to affect either or both diabetes forms.
- There is no mentioning/references in the main text to the figures. Please refer to the figures within the main text.
- I was wondering if there is any link between miRNAs and genetic risk variants? Do any of the mentioned miRNAs harbour variants that change the sequence of the miRNAs or change their expression levels in beta-cells? I would consider such information highly relevant to include in the manuscript.
- In line 141, the 'an' should be changed to 'a'.
Author Response
We thank both the reviewers for the constructive and positive comments and observations, that of course will render the review easier to read and will highlight both the solved as well as the critical aspects of beta cell dedifferentiation. All the provided modifications are shown in red.
The term 'temporary' used in line 38 may mistakingly lead to reader to think that the loss of beta-cell function will at some point return to normal, but this is clearly not the case as the dedifferrentiation or dysfunction of these 'silent' beta-cells seem chronic. Consider to rephrase the term. We modified the sentences: the terms ‘definitive’ and ‘temporary’ have been removed, just describing the two main mechanisms leading to the loss of functional beta cell mass.
In general, the authors do not differentiate between which miRNAs that they describe are mainly relevant for type 1 diabetes and which for type 2 diabetes and which are for both. Maybe it would strenghten the manuscript if the authors put slightly more attention as to describe which specific miRNAs are believed to affect either or both diabetes forms. We specified the diabetes context, where necessary.
There is no mentioning/references in the main text to the figures. Please refer to the figures within the main text. We mentioned all figures in the main text.
I was wondering if there is any link between miRNAs and genetic risk variants? Do any of the mentioned miRNAs harbour variants that change the sequence of the miRNAs or change their expression levels in beta-cells? I would consider such information highly relevant to include in the manuscript. Among the described beta-cell essential miRNAs, only miR-375 has been reported (By Ciccacci et al. 2013; DOI 10.1007/s00592-013-0469-7) to show a particular SNP rs6715345 C->G; however, this SNP was not more incident in T2D vs non-diabetic subjects, thus being inconclusive from a genetic point of view. As an example, in the same study, the authors reported that the G allele of rs531564 in hsa-mir-124a, is more frequent in T2D than in controls and, consequently, behaves like a risk allele (OR = 2.15, P = 0.008). Most importantly, the rs531564 polymorphism in the pri-miRNA influences the expression of miR-124; in details, Northern blot and real-time PCR analysis showed that the amount of mature miR-124 from the C/G heterozygosity of rs531564 was increased compared with the CC or GG homozygosity of rs531564 in Alzheimer’s disease, although this result has never been confirmed in beta cells or, in general, in pancreas. Nevertheless, we added 3 references on the importance of miR-124 in pancreas development, that, until now, has not been linked to beta-cell dedifferentiation and, for this reason, was not deepened in our review.
In line 141, the 'an' should be changed to 'a'. The word has been modified.

Reviewer 2 Report
In this review, Grieco and collaborators describe most of the relevant aspects of miRNAs in relation to beta cell differentiation, maintenance and function.
The introduction to miRNAs is clear and concise, however I have found the section on “beta cell identity and function in health and disease” a little bit too long and complex. I think that cutting back this section would improve the equilibrium between different sections and highlight the most important part of the review which actually starts in Section 3.
In relation to the content, I think that the review is up-to-date. I see no flaws or important missing information, although I would like the authors to develop a little further the potential therapeutic use of miRNA modulators or their potential as biomarkers of beta cell function.
In addition, I would advise the authors to double check the English style. Throughout the text there are some grammatical errors and wrong expressions which might make it difficult for readers to understand the information presented. In addition, there are a number of paragraphs for which it is difficult to find a clear meaning (for example, page 10, lines 430-440; page 9, lines 388-391; page 10, lines 415-419; page 12, lines 515-523). I suggest to have the manuscript revised by a native English speaker.
Finally, I would like the authors to consider the following minor points:
- Please use inclusive language, replace “man” by “human” (lines 232 and 381)
- Please put in italics the name of genes or RNAs when needed.
- Please separate “ßcell” (found throughout the manuscript, starting in Keywords section)
- Page 3, line 115: specify “GSIS”
- Page 4, line 141: replace “an high” for “a high”
- Page 4, line 164: remove “dependent on the presence of different energy substrates”
- Page 7, lines 313 (and so on): change “miR-375KO” by “miR-375 KO” or “miR-375-KO” (the same for miR-204KO)
- Page 9, Legend for Figure 2: line 396, remove “in turn” and line 398, replace “resulting into improvement” with “resulting into an improvement”
- Page 10, line 22: correct “ER-relayed”
- Page 11, line 455: please remove the question and rephrase to introduce this point.
- Page 11, line 459: remove “major direct or indirect”
- Page 12, lines 486-490: This statement needs to be further elaborated as it is unclear what exactly the point the authors are trying to convey.
- Page 14, line 603: remove “in” before T1D and T2D.
Author Response
We thank both the reviewers for the constructive and positive comments and observations, that of course will render the review easier to read and will highlight both the solved as well as the critical aspects of beta cell dedifferentiation. All the provided modifications are shown in red.
In this review, Grieco and collaborators describe most of the relevant aspects of miRNAs in relation to beta cell differentiation, maintenance, and function.
The introduction to miRNAs is clear and concise, however I have found the section on “beta cell identity and function in health and disease” a little bit too long and complex. I think that cutting back this section would improve the equilibrium between different sections and highlight the most important part of the review which actually starts in Section 3. We fully agree with reviewer’s comment. Indeed, we shortened the section “beta cell identity and function in health and disease” in order to highlight the most important aspects of our review.
In relation to the content, I think that the review is up-to-date. I see no flaws or important missing information, although I would like the authors to develop a little further the potential therapeutic use of miRNA modulators or their potential as biomarkers of beta cell function. One informative sentence as well as several critical comments on the therapeutic use of miRNA-based drugs have been added in the paragraph.
In addition, I would advise the authors to double check the English style. Throughout the text there are some grammatical errors and wrong expressions which might make it difficult for readers to understand the information presented. In addition, there are a number of paragraphs for which it is difficult to find a clear meaning (for example, page 10, lines 430-440; page 9, lines 388-391; page 10, lines 415-419; page 12, lines 515-523). I suggest to have the manuscript revised by a native English speaker.
English style has been revised and all minor points of corrections have been modified.
Finally, I would like the authors to consider the following minor points:
- Please use inclusive language, replace “man” by “human” (lines 232 and 381)
- Please put in italics the name of genes or RNAs when needed.
- Please separate “ßcell” (found throughout the manuscript, starting in Keywords section)
- Page 3, line 115: specify “GSIS”
- Page 4, line 141: replace “an high” for “a high”
- Page 4, line 164: remove “dependent on the presence of different energy substrates”
- Page 7, lines 313 (and so on): change “miR-375KO” by “miR-375 KO” or “miR-375-KO” (the same for miR-204KO)
- Page 9, Legend for Figure 2: line 396, remove “in turn” and line 398, replace “resulting into improvement” with “resulting into an improvement”
- Page 10, line 22: correct “ER-relayed”
- Page 11, line 455: please remove the question and rephrase to introduce this point.
- Page 11, line 459: remove “major direct or indirect”
- Page 12, lines 486-490: This statement needs to be further elaborated as it is unclear what exactly the point the authors are trying to convey. The paragraph on miR-338 has been removed in the light of the weaker evidence available on the role of this microRNA, compared to other miRNAs.
- Page 14, line 603: remove “in” before T1D and T2D.
